# Are Further Interventions Needed to Prevent and Manage Hospital-Acquired Hyponatraemia? A Nationwide Cross-Sectional Survey of IV Fluid Prescribing Practices

**DOI:** 10.3390/jcm9092790

**Published:** 2020-08-29

**Authors:** Per Sindahl, Christian Overgaard-Steensen, Helle Wallach-Kildemoes, Marie Louise De Bruin, Hubert GM Leufkens, Kaare Kemp, Helga Gardarsdottir

**Affiliations:** 1Danish Medicines Agency, Division of Pharmacovigilance and Medical Devices, 2300 Copenhagen, Denmark; kake@dkma.dk; 2Division of Pharmacoepidemiology and Clinical Pharmacology, Utrecht Institute for Pharmaceutical Sciences, Faculty of Science, Utrecht University, 3584CG Utrecht, The Netherlands; H.G.M.Leufkens@uu.nl (H.G.L.); H.Gardarsdottir@uu.nl (H.G.); 3Copenhagen Centre for Regulatory Science, Department of Pharmacy, Faculty of Health and Medical Sciences, University of Copenhagen, 2100 Copenhagen, Denmark; marieke.debruin@sund.ku.dk,; 4Department of Intensive Care 4131, Rigshospitalet, 2100 Copenhagen, Denmark; christian.overgaard.steensen@regionh.dk; 5Section for Social and Clinical Pharmacy, Department of Pharmacy, Faculty of Health and Medical Sciences, University of Copenhagen, 2100 Copenhagen, Denmark; helle.wallach@sund.ku.dk; 6Department of Clinical Pharmacy, Division Laboratories, Pharmacy and Biomedical Genetics, University Medical Center Utrecht, 3584CX Utrecht, The Netherlands

**Keywords:** hyponatraemia, fluid therapy, intravenous fluids, prescribing practice, knowledge

## Abstract

Background: Hyponatraemia is associated with increased morbidity, increased mortality and is frequently hospital-acquired due to inappropriate administration of hypotonic fluids. Despite several attempts to minimise the risk, knowledge is lacking as to whether inappropriate prescribing practice continues to be a concern. Methods: A cross-sectional survey was performed in Danish emergency department physicians in spring 2019. Prescribing practices were assessed by means of four clinical scenarios commonly encountered in the emergency department. Thirteen multiple-choice questions were used to measure knowledge. Results: 201 physicians responded corresponding to 55.4% of the total population of physicians working at emergency departments in Denmark. About a quarter reported that they would use hypotonic fluids in patients with increased intracranial pressure and 29.4% would use hypotonic maintenance fluids in children, both of which are against guideline recommendations. Also, 29.4% selected the correct fluid, a 3% hypertonic saline solution, for a patient with hyponatraemia and severe neurological symptoms, which is a medical emergency. Most physicians were unaware of the impact of hypotonic fluids on plasma sodium in acutely ill patients. Conclusion: Inappropriate prescribing practices and limited knowledge of a large number of physicians calls for further interventions to minimise the risk of hospital-acquired hyponatraemia.

## 1. Introduction

Hyponatraemia, defined as a plasma sodium (P-Na) < 135 mmol/L, is the most common electrolyte disorder affecting 15% to 30% of hospitalised patients, and is associated with increased mortality, morbidity, and length of hospital stay [1]. The aetiology of hyponatraemia is multifactorial, but is frequently hospital-acquired due to the administration of hypotonic intravenous fluids [1]. Hoorn and colleagues found that ten percent of children admitted to the emergency department (ED) developed acute hyponatraemia due to hypotonic fluid administration [2]. Worldwide, more than 100 cases of death or permanent brain damage have been reported resulting from hospital-acquired hyponatraemia after administration of hypotonic intravenous fluids. Many of these involved otherwise healthy adults and children [1].

In healthy individuals, the body maintains P-Na within the normal range (135 to 145 mmol/L) by controlling water intake (thirst/nausea) and excretion [3]. However, acutely ill patients are at high risk of developing non-osmotic antidiuretic hormone (ADH) secretion leading to water retention [1]. In addition, the administration of intravenous fluids bypasses normal protective mechanisms of thirst and nausea. The combination of hypotonic intravenous fluids and water retention leads to an excess of water relative to sodium and puts the patient at risk of hyponatraemia. Rapid reduction in P-Na makes cells swell by promoting water movement from the extracellular fluid into the cells, which may cause cerebral symptoms (hyponatraemic encephalopathy) due to brain oedema. If not addressed acutely, this might evolve to brain damage and death. On the other hand, rapid and excessive correction of P-Na can also lead to brain injury due to osmotic demyelination syndrome (ODS) [4].

Fortunately, hospital-acquired hyponatraemia is largely preventable by appropriate use of intravenous fluids as first suggested by Moritz and Ayus in 2003 for maintenance fluids in children [5]. Since then, a growing body of evidence has shown the association between maintenance hypotonic fluids with subsequent development of hyponatraemia in children [6,7,8,9,10]. Furthermore, if hyponatraemia occurs, early recognition of severe symptoms and appropriate management might prevent its severe complications [1,11,12].

Several attempts have been made by medical associations and national regulatory authorities to reduce the risk of hospital-acquired hyponatraemia caused by intravenous fluids [1,13,14,15]. Lately in 2018, the American Academy of Pediatrics published a guideline on the use of maintenance intravenous fluid therapy in children after concerns of iatrogenic hyponatraemia were raised [16]. In Europa, the European Pharmacovigilance Risk Assessment Committee concluded in July 2017 that hospital-acquired hyponatraemia in association with hypotonic fluids continues to be an important risk, despite previous attempts to reduce that risk. Subsequently, the European product information of hypotonic intravenous fluids containing electrolytes or carbohydrates has been updated to include warnings about hospital-acquired hyponatraemia. Yet for some risks, this will not be sufficient and additional risk minimisation measures adapted to the national needs may be necessary to manage the risk [17].

The primary aim of this study was to explore intravenous fluid prescribing practices among physicians working at EDs in Denmark, in order to determine whether further interventions are needed to reduce the risk of hospital-acquired hyponatraemia.

## 2. Methods

### 2.1. Setting and Study Population

We conducted a cross-sectional survey amongst physicians working at EDs throughout Denmark using a self-administered questionnaire (the questionnaire is provided in Appendix A). We focused on EDs because intravenous fluid prescriptions are initiated in the ED, and previous research have shown that errors in prescription of intravenous fluids are particularly likely in EDs [18]. The recruitment started in March 2019 and ended in May that year.

### 2.2. Recruitment

We invited physicians from all 38 EDs in Denmark, distributed over 21 hospitals, by mail to participate in the survey. In case of no response to the first invitation, two more reminder attempts were made. In order to increase the response rate, we offered two options to fill out the questionnaire. In agreement with the head of the department, the questionnaire was either distributed via an online link by the head of the department, or a paper version was distributed in person and completed during the daily meeting.

Participation was anonymous and voluntary, and consent was not necessary according to the Danish Data Protection Authority.

### 2.3. Development of the Questionnaire

The objective of this study was to measure intravenous fluid prescribing practices and knowledge pertaining to hyponatraemia and intravenous fluids. A questionnaire was developed and reviewed by a team of experts including an intensive care physician with extensive experience from the ED and expert in intravenous fluid treatment, pharmacovigilance officers, and researchers in social pharmacy, pharmacoepidemiology, and regulatory science.

In addition, cognitive pretesting was performed in four physicians according to the ‘Thinking aloud method’ and ‘Individual debriefing method’ [19]. Following the pre-test and review, minor changes were made to reduce the number of questions, and to increase the clarity and understanding of each question. Pre-testing evidenced that it would take approximately 20 min to complete the questionnaire.

### 2.4. Outcome Measures

Four clinical scenarios (hereafter called scenarios) describing different situations encountered in the ED where IV fluids are administered were used to measure prescribing practices of intravenous fluids [20,21]. The scenarios selected for this survey all represent conditions associated with increased ADH secretion where the risk of developing hyponatraemia has been well documented [1]. The scenarios covered:A high-risk (potentially increased intracranial pressure) patient with hypovolaemia.A child in need of maintenance intravenous fluids without hypovolaemia and hyponatraemia.A hypovolaemic and hyponatraemic (P-Na = 110 mmol/L) patient without severe symptoms of hyponatraemia.A hyponatraemic (P-Na = 118 mmol/L) patient with severe symptoms of hyponatraemia.

After each scenario, participants were asked to select the first-line treatment of choice between eight commonly used intravenous fluids with different electrolytes and/or carbohydrates content and tonicity (see Table 1). In all scenarios it was noted that the patient was not tolerating any oral intake.

The primary outcome measures were prescribing practice measured as correct answers to the scenario questions and selection of hypotonic fluids as this is associated with hospital-acquired hyponatraemia.

After the four scenarios, participants were asked to answer 13 factual knowledge questions within the following topics:Renal water excretion in the acutely ill patient.Intravenous fluids impact on P-Na in the acutely ill patient.Hyperglycaemia and P-Na.Severe symptoms of hyponatraemia.Patients at high risk of severe symptoms.Prevention and treatment of over-correction of hyponatraemia.

### 2.5. Data Analysis

We used LimeSurvey version 2.67.2 (Limesurvey GmbH. / LimeSurvey: An Open Source survey tool /LimeSurvey GmbH, Hamburg, Germany. http://www.limesurvey.org) for the electronic version of the questionnaire.

All questions were summarized using descriptive statistics (counts and percentages) of correct responses. For the scenarios, also selection of hypotonic fluids was summarized (counts and percentages).

Simple logistic regression was used to explore predictors of prescribing practice. The aim of this analysis was to decide whether we should target the interventions at certain EDs or towards certain groups of physicians. Based on responses to the four scenarios, we defined a priori, critical prescribing practice as comprising of none or one correct response, and excellent prescribing practice equivalent to three or four correct responses. Excellent prescribing practice indicates no need for additional risk minimisation measures, and a critical prescribing practice indicates a need for an intensified and differentiated intervention. The following predictors were assessed: age, number of weekly treated patients with intravenous fluids, position, years of practice, region, size of the hospital, complexity of services they provide, and type of patients they serve.

All data handing and analysis was performed using SPSS (IBM Corp. Released 2017. IBM SPSS Statistics for Windows, Version 25.0. Armonk, NY, USA.).

## 3. Results

### 3.1. Characteristics of Respondents

Fifteen (71%) out of 21 hospitals, and 23 (61%) out of the 38 invited EDs participated in the study. Four (11%) EDs refused, and 11 (29%) did not respond to the invitation.

The primary analysis is based on the 201 respondents who responded to all scenarios, representing 55.4% (201/363) of the total population of physicians working at EDs in Denmark based on the estimated source population from 2014 (see details of breakoff and study population in Figure 1) [22].

Overall, respondents were experienced, 52.6% treated more than five patients a week with intravenous fluids, 59.1% had more than five years of practice, and 43.7% were consultants. No physicians from EDs of low complexity participated. Of note, characteristics of respondents were similar distributed among the three subsets of respondents (Table 2).

### 3.2. Response to Scenario and Knowledge Questions

Of the 201 participants who responded to all scenarios, 1.5% (3/201) answered all four questions correctly, 5.0% (10/201) three, 59.2% (119/201) two, 27.9% (56/201) one, and 6.5% (13/201) answered none correctly. In terms of excellent and critical prescribing practices, this corresponds to 6.5% and 34.4%, respectively. The fluids selected for each scenario are detailed in Table 3.

In scenario 2 and 3, hypotonic fluids included the strongly hypotonic fluids (see Table 1). In scenario 1 and 4, hypotonic fluids also included the moderately hypotonic fluid, Ringer’s acetate. With a slightly lower sodium concentration than the physiologic range in plasma (130 mmol/L versus 135 to 145 mmol/L in plasma), Ringer’s acetate is hypotonic, and larger amounts can drive sodium levels to hyponatraemic ranges [23], which is critical in scenario 1 and 4.

In the first scenario, describing normo-natraemic and hypovolaemic women with potentially increased intracranial pressure (meningitis), 71.6% (144/201) choose the correct intravenous fluid (isotonic saline solution) to restore normal hydration status. Hypotonic solutions including the moderate hypotonic solution, Ringer’s acetate, were incorrectly selected by 24.9% (50/201). Due to the low sodium content, hypotonic fluids are inappropriate to restore the circulatory volume. Therefore, isotonic saline solution and Ringer’s acetate/lactate are the fluids of choice for most patients with hypovolaemia [24,25]. However, in patients with pre-existing increased intracranial pressure (e.g., meningitis, cerebral contusion, and acute liver failure), even a small decrease in P-Na induced by slightly hypotonic fluids like Ringer’s acetate/lactate can increase intracranial pressure critically [26].

The second scenario presented a 5-year-old boy in need of maintenance intravenous fluid. The correct option (isotonic saline solution with 5% glucose) was selected by 10.4% (21/201). The most commonly selected fluid (83 out of 201 corresponding to 41.3%), though incorrect, was the isotonic saline solution. Hypotonic intravenous fluids were incorrectly selected by 29.4% (59/201).

The third scenario was a 75-year-old hypovolaemic and hyponatraemic (P-Na = 110 mmol/L) woman in treatment with thiazide diuretics without severe symptoms of hyponatraemia. The correct fluid (either Ringer’s acetate or isotonic saline solution) was selected by 67.7% (136/201), and 10.4% (21/201) selected hypotonic intravenous fluids incorrectly.

The fourth scenario was a 28-year-old hyponatraemic (P-Na = 118 mmol/L) woman with severe symptoms of hyponatraemia (polydipsia, altered level of consciousness, and vomiting) [4,27,28,29]. The correct option (hypertonic saline solution) was selected by 29.4% (59/201). The most commonly selected fluid, though incorrect, was the isotonic saline solution (76 out of 201 corresponding to 37.8%). Hypotonic intravenous fluids were incorrectly selected by 16.5% (33/201).

Percentage of correct responses to the knowledge questions are listed in Table 4 below. The median number of correct responses of the 159 respondents who answered all knowledge questions was 5/13 (interquartile range: 3/13 to 6/13).

### 3.3. Analyses of Prescribing Practice by Demographical Variables and Characteristics of EDs

The predictive value of characteristics of respondents and EDs on critical prescribing practice is presented in Table 5. Since few respondents (13) performed excellently, the regression analysis was only applied with a cut-off value of less than two (defined a priori as critical prescribing practice) for the outcome variable. The regression analysis showed that none of the characteristics were significantly associated with critical prescribing practice.

## 4. Discussion

### 4.1. Main Findings

The most striking finding of this study was the choice of hypotonic fluids in high-risk patients; a quarter of respondents selected a hypotonic fluid for a patient with potentially increased intracranial pressure and more than a quarter selected a hypotonic maintenance fluid for a child. Moreover, if a patient develops hyponatraemia during hospitalisation due to intravenous fluid treatment, it would probably not be linked to the administration of intravenous fluids, since most physicians were unaware of the impact of hypotonic fluids on P-Na in acutely ill patients. Finally, even if the respondents were able to link the treatment of intravenous fluids to hyponatraemia, less than one-third knew how to treat a patient with severe symptoms of hyponatraemia, which is a medical emergency.

#### 4.1.1. Use of Hypotonic Intravenous Fluids

Children and patients with potentially increased intracranial pressure are patient-populations at particular risk of hyponatraemic encephalopathy upon hypotonic intravenous fluid treatment. Consequently, the European product information of physiologically hypotonic intravenous fluids was updated in 2017/2018 to include, amongst others, a warning of hospital-acquired hyponatraemia in these patients [17]. In addition, treatment with hypotonic fluids (including Ringer’s acetate/lactate) are against guideline recommendations in patients with potentially increased intracranial pressure [17,26,27,30], and in children for routine maintenance requirements [15,16,31]. Despite these warnings, our study indicates that hypotonic fluids continue to be used in high-risk patients. The selection of hypotonic fluids (24.9%) in scenario 1, covering a patient with potentially increased intracranial pressure, was mainly driven by Ringer’s acetate, and which was selected by 13.9%. This result may be due to Ringer’s acetate being termed as “balanced” or “physiologic” solutions. However, Ringer’s acetate is neither truly balanced nor physiologic [23].

Numerous studies of maintenance intravenous fluids in children have shown that isotonic fluids are effective in preventing hospital-acquired hyponatraemia [1]. Furthermore, there is no evidence of increased adverse effects such as hypernatraemia, hyperchloraemic metabolic acidosis, and fluid overload with isotonic maintenance fluids [16]. Yet it remains uncertain how many patients would need to be treated with isotonic fluids to prevent a rare, but potentially devastating event like hyponatraemic encephalopathy [32].

Of note in scenario 2, describing a child in need of maintenance fluids, physicians working at a paediatric ED performed better than physicians working at adult EDs when comparing those correctly answered (isotonic NaCl with 5% glucose); 16% versus 4%, respectively. However, when looking at selection of hypotonic fluids, physicians working at a paediatric ED performed more poorly; 40% selected a hypotonic fluid as opposed to 17% by physicians working at adult EDs.

Our results are consistent with both a recent study conducted in Spain which reveals that 29% of all paediatricians there continue to use hypotonic fluids as maintenance intravenous fluid [33]; as well as a study conducted in the USA showing that the majority of paediatricians choose to use hypotonic maintenance intravenous fluid in daily practice [34]. Considering that intravenous fluids are one of the most commonly prescribed therapies [35], our results on prescribing practices in children and patients with potentially increased intracranial pressure support further risk minimisation measures.

#### 4.1.2. Linking Hypotonic Intravenous Fluid Treatment and Hyponatraemia

Identifying hyponatraemia from mismanagement of intravenous fluid therapy and establishing a causal association between the two can be difficult. Our study may shed light on this challenge. In the knowledge questions for example, participants were asked about intravenous fluids’ impact on P-Na in patients with reduced renal water excretion. Since only 24.0% to 46.2% was able to classify the selected and commonly used intravenous fluids correctly, our findings suggest a lack of knowledge about tonicity and its physiological effects, which may preclude recognition of the association between hypotonic fluids and hyponatraemia.

In the clinic, linking hypotonic intravenous fluid treatment and hyponatraemia is further challenged. Firstly, differing personnel are involved in taking care of patients; one person might set up the intravenous fluid, but another may later inspect the patient and assess the treatment. Secondly, the presenting symptoms (e.g., headache, nausea, vomiting, and weakness) and, to some extent the severe symptoms (e.g., altered level of consciousness), are nonspecific and easily overlooked, which could delay a diagnosis. The length of time separating the administration and a diagnosis complicates and hinders the recognition of hyponatraemia and its association to intravenous fluid treatment. Confused labelling, whereby many strong hypotonic fluids (e.g., 5% Glucose) are labelled as ‘isotonic’ though become physiologically hypotonic once the glucose is metabolized, challenges the issue further.

Our findings are in accordance with the NICE guideline that states that there is a general lack of knowledge amongst health professionals concerning the composition of intravenous fluids and that mismanagement is rarely reported [18].

#### 4.1.3. Treatment of Hyponatraemia with Severe Symptoms

Although isotonic maintenance fluids reduce the risk of hospital-acquired hyponatraemia, it will not eliminate the problem completely as hospital-acquired hyponatraemia is common in children receiving isotonic fluids as well [36]. In addition, sodium is often low in both children and adults on admission [37,38]. Hyponatraemia should therefore always be considered a cause of neurological symptoms in hospitalised patients regardless of the IV treatment. Since occurrence of severe symptoms of hyponatraemia is a medical emergency requiring immediate treatment to prevent neurological sequelae and death, it is crucial that physicians are able to recognise symptoms of hyponatraemia. Treatment of hyponatraemia should be based on neurological symptoms, not the absolute plasma sodium concentration [4], and managed with 3% NaCl [4,27,28,29]. In our study, less than a third of the respondents selected the correct fluid for a hyponatraemic patient with severe neurological symptoms. When participants were asked to mark severe symptoms of hyponatraemia requiring acute treatment from a list of six options including three distractors, less than half of the respondents answered all six correctly. Consistent with studies conducted in UK, Spain, and the Netherlands, our results suggest that severe symptoms may not be recognised and treated properly [39,40,41]. In contrast, an Australian study showed that hypertonic (3%) saline was appropriately used in cases of severe symptomatic hyponatraemia [42].

### 4.2. Strengths and Limitations

The results of this study should be presented in perspective to its strengths and limitations. The strength of our study is that we were able to approach the complete source population. In addition, the scenarios were based on real-life cases and developed in close collaboration with a clinician, and expert within the field of intravenous fluid treatment.

There were several limitations to this study. First, we cannot conclude whether inappropriate prescribing practice contributed to adverse patient outcomes. However, it was not considered feasible to measure actual patient outcomes in our study due to lack of intravenous fluid administration recording and the underreporting of complications to intravenous fluid therapy. A scenario-based survey was therefore considered a more valid method to measure prescribing practice [20,21].

Secondly, with regard to prescribing practice and knowledge, we do not know how those who agreed to participate differ from those who refused or did not respond to the invitation. Together with a response rate on 55.4%, this clearly limits the generalizability of our study. However, it is worth noticing that the response rate represents 55.4% of all physicians working at an ED in Denmark, and there was a balanced distribution with regards to region, type of hospital, and experience, with a slight overweight of experienced physicians as consultants accounted for 44% of our study compared to 35% in the population [23]. Therefore, we believe that the study population is likely to be representative of physicians working at EDs in Denmark. From a broader perspective, the single-country setting may limit extrapolation to other countries depending on the specific healthcare system and clinical practice. One factor that might have influenced the response rate, is the length of the questionnaire. As can be seen in Figure 1, break-off-rate increases with length of the questionnaire. Furthermore, an estimated completion time of 20 min might have influenced the willingness of the ED managers to participate in the study.

Lastly, there is uncertainty and controversy about whether scenarios reflect actual clinical practice. The Hawthorne effect, a tendency to perform better when observed, seems inevitable [21]. In addition, the scenario setting might not reflect everyday clinical practice, where practitioners are busy, and get distracted under clinical pressure. Moreover, there might have been a selection bias towards those for whom the questions were manageable. All together, these biases tend to over-estimate the performance of the physicians.

### 4.3. Clinical Implications

Our findings support the recommendations by NICE and several other studies regarding the need for guidance on intravenous fluid therapy [18,43,44].

It is important to recognise that intravenous fluids are medicines and intravenous fluid prescribing should be held in the same regard to that of any medicines in order to avoid serious complications from intravenous fluid treatment. Thus, regular monitoring is needed so that responses to intravenous fluid treatment can be evaluated and altered or stopped as appropriate. When monitoring P-Na, it is important to use the same technique as differing techniques can yield different results [45].

This study indicates that inappropriate prescribing practices may be caused by insufficient knowledge, which can help with tailoring the needed intervention. Based on our findings, the following areas may require additional attention when setting up an educational program for prescribers:The association between hyponatraemia and hypotonic intravenous fluids.Intravenous fluids’ impact on P-Na in patients with reduced water excretion.Treatment of patients with severe symptoms of hyponatraemia.

In our regression analysis, we observed no significant differences in prescribing practices between senior and junior doctors and the other selected characteristics, hence there is no reason to differentiate or target the intervention at certain EDs or towards certain groups of physicians.

In conclusion, the current prescribing practices of a large numbers of physicians working at EDs in Denmark calls for further interventions to reduce the risk of hospital-acquired hyponatraemia.

## Figures and Tables

**Figure 1 jcm-09-02790-f001:**
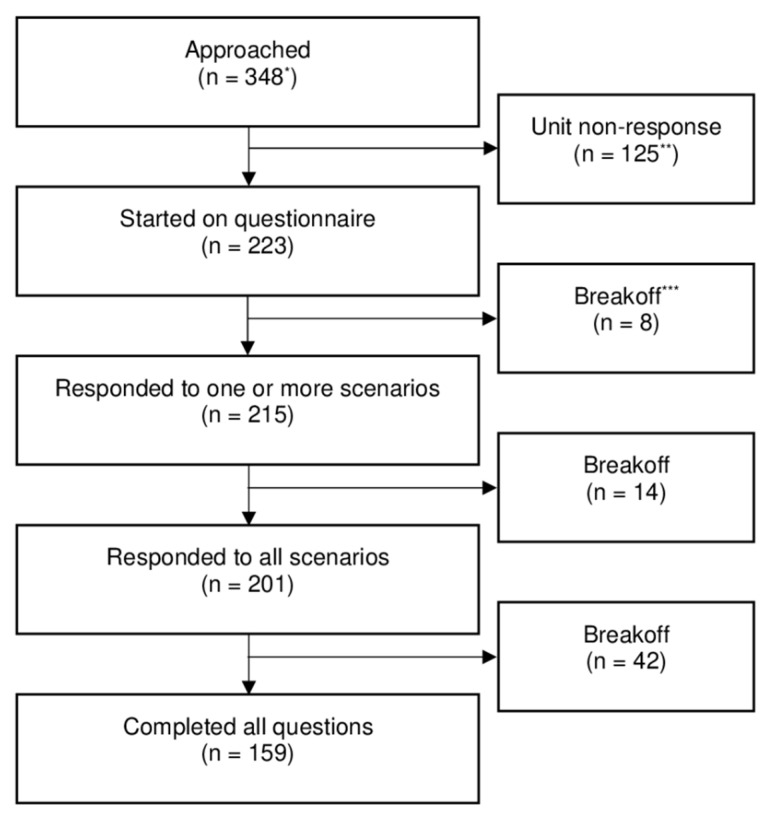
Consort diagram of participants. * Estimated source population (i.e., physicians working at emergency departments in Denmark) from 2014. ** This number includes physicians at the four EDs that refused, the 11 EDs that did not respond to the invitation, and physicians at participating EDs who choose not to participate. *** Respondents started on the questionnaire but failed to complete it resulting in breakoff.

**Table 1 jcm-09-02790-t001:** IV fluid response choices indicating sodium concentration and tonicity after injection.

Fluid Response Choices of the Scenarios *	Sodium Concentration	Tonicity After Injection
Glucose 5% isotonic	0 mmol/L	Strongly hypotonic
Darrow-glucose	31 mmol/L	Strongly hypotonic
Potassium-sodium-glucose	40 mmol/L	Strongly hypotonic
0.45% sodium chloride with 2.5% glucose isotonic	77 mmol/L	Strongly hypotonic
Ringer’s acetate	130 mmol/L	Moderately hypotonic
Isotonic saline solution	154 mmol/L	Isotonic
0.9% NaCl with 5% glucose	154 mmol/L	Isotonic
3% NaCl	513 mmol/L	Strongly hypertonic

* see Appendix A for a full description of the composition of the fluids.

**Table 2 jcm-09-02790-t002:** Characteristics and distribution of physicians indicating gender, age, number of weekly treated patients with intravenous fluids, years of practice and position, and characteristics of hospitals/emergency departments (EDs) indicating location (region), size, complexity, and patient type served.

	Responded to One or More Scenarios		Responded to All Scenarios		Completed All Questions	
	*n*	(%)	*n*	(%)	*n*	(%)
All	215	(100.0)	201	(93.4)	159	(74.0)
**Characteristics of respondents**						
Gender						
Female	126	(58.6)	119	(59.2)	88	(55.3)
Male	89	(41.4)	82	(40.8)	71	(44.7)
Age						
18–34 years	88	(40.9)	80	(39.8)	64	(40.3)
35–44 years	57	(26.5)	54	(26.9)	43	(27.0)
≥45 years	70	(32.6)	67	(33.3)	52	(32.7)
Number of weekly treated patients with intravenous fluids						
0 patients	29	(13.5)	28	(13.9)	21	(13.2)
1–5 patients	72	(33.5)	68	(33.8)	58	(36.5)
>5 patients	113	(52.6)	104	(51.7)	80	(50.3)
Unknown	1	(0.5)	1	(0.5)	0	(0)
Years of practice						
≤5 years	86	(40.0)	78	(38.8)	61	(38.4)
>5 years	127	(59.1)	121	(60.2)	97	(61.0)
Unknown	2	(0.9)	2	(1.0)	1	(0.6)
Position						
Junior doctor	121	(56.3)	111	(55.2)	89	(56.0)
FY1 *	30	(14.0)	25	(12.4)	18	(11.3)
FY2 **	27	(12.6)	26	(12.9)	23	(14.5)
Specialty registrar	50	(23.3)	47	(23.4)	39	(24.5)
Other ***	14	(6.5)	13	(6.5)	9	(5.7)
Senior doctor (Consultant)	94	(43.7)	90	(44.8)	70	(44.0)
**Characteristics of EDs** ****						
Size						
Large	84	(39.1)	79	(39.3)	61	(38.4)
Medium	111	(51.6)	104	(51.7)	85	(53.5)
Small	20	(9.3)	18	(9.0)	13	(8.2)
Complexity						
High	65	(30.2)	62	(30.8)	44	(27.7)
Medium	150	(69.8)	139	(69.2)	115	(72.3)
Low	0	(0)	0	(0)	0	(0)
Type						
Combined general population ED *****	31	(14.4)	29	(14.4)	23	(14.5)
Adult ED	76	(35.3)	69	(34.3)	52	(32.7)
Pediatric ED	106	(49.3)	101	(50.2)	82	(51.6)
Trauma center	2	(0.9)	2	(1.0)	2	(1.3)

* Foundation doctor year 1. ** Foundation doctor year 2. *** ‘Other’ includes medical students (7), unspecified junior doctor (4), pre FY1 (2), and a PhD student. **** Characteristics of the EDs was based on a report of EDs by the Danish Ministry of Health [22]. ***** Combined general population EDs provide care for all patients in one area, while separate general population EDs provide care to children and adults in separate locations within a facility.

**Table 3 jcm-09-02790-t003:** Percentage of correct responses of prescribing practice in four clinical scenarios encountered in the emergency department. Responses are grouped into four categories, selection of correct fluid(s), inappropriate fluid(s), incorrect hypotonic fluid(s) and ‘do not know’.

	*n*	(%)
Scenario 1 (*n* = 201) An otherwise healthy 18-year-old girl is hospitalised on suspicion of meningitis. She has thrown up and has diarrhoea. On examination, she appears pale with cold skin, a slightly increased heart rate, normal blood pressure, and with decreased level of consciousness (Glasgow Coma Scale = score 14). Laboratory tests are normal.		
**Correct fluid**		
Isotonic saline, [Na^+^] = 154 mmol/L	144	(76.1)
**Inappropriate fluids**		
0.9% NaCl with 5% glucose, [Na^+^] = 154 mmol/L	5	(2.5)
3% NaCl, [Na^+^] = 513 mmol/L	0	(0)
**Incorrect hypotonic fluids**		
Glucose 5% isotonic	1	(0.5)
Darrow-glucose, [Na^+^] = 31 mmol/L	1	(0.5)
Potassium-sodium-glucose, [Na^+^] = 40 mmol/L	14	(7.0)
0.45% sodium chloride with 2.5% glucose isotonic, [Na^+^] = 77 mmol/L	6	(3.0)
Ringer’s acetate, [Na^+^] = 130 mmol/L	28	(13.9)
**Do not know**	2	(1.0)
Scenario 2 (*n* = 201) A 5-year-old boy arrives at the emergency department with a head injury after falling from a bike. He has headache and nausea, but no vomiting or signs of hypovolaemia. He has been unconscious for half an hour; however, the CT scan, clinical examination and laboratory results are all normal.		
**Correct fluid**		
0.9% NaCl with 5% glucose, [Na^+^] = 154 mmol/L	21	(10.4)
**Inappropriate fluid**		
Isotonic saline, [Na^+^] = 154 mmol/L	83	(41.3)
Ringer’s acetate, [Na^+^] = 130 mmol/L	22	(10.9)
3% NaCl, [Na^+^] = 513 mmol/L	1	(0.5)
**Incorrect hypotonic fluids**		
Glucose 5% isotonic	3	(1.5)
Darrow-glucose, [Na^+^] = 31 mmol/L	6	(3.0)
Potassium-sodium-glucose, [Na^+^] = 40 mmol/L	42	(20.9)
0.45% sodium chloride with 2.5% glucose isotonic, [Na^+^] = 77 mmol/L	8	(4.0)
**Do not know**	15	(7.5)
Scenario 3 (*n* = 201) A 75-year-old woman arrives at the emergency department with hip fracture after a fall. There are no signs of head injury. The patient has had a poor appetite for a long time. Medical history includes thiazide diuretics for hypertension, but otherwise she is healthy. Clinical examination shows symptoms of hypovolaemia: cold and pale skin, heart rate at 100 bpm, and a slightly increased respiratory rate. Laboratory tests show P-Na = 110 mmol/L.		
**Correct fluids**		
Isotonic saline, [Na^+^] = 154 mmol/L	120	(59.7)
Ringer’s acetate, [Na^+^] = 130 mmol/L	16	(8.0)
**Inappropriate fluids**		
0.9% NaCl with 5% glucose, [Na^+^] = 154 mmol/L	11	(5.5)
3% NaCl, [Na^+^] = 513 mmol/L	16	(8.0)
**Incorrect hypotonic fluids**		
Glucose 5% isotonic	4	(2.0)
Darrow-glucose, [Na^+^] = 31 mmol/L	0	(0)
Potassium-sodium-glucose, [Na^+^] = 40 mmol/L	10	(5.0)
0.45% sodium chloride with 2.5% glucose isotonic, [Na^+^] = 77 mmol/L	7	(3.5)
**Do not know**	17	(8.5)
Scenario 4 (n = 201) A 28-year-old woman is hospitalised on suspicion of medication poisoning and large intake of water. She vomits and complains about headaches. She exhibits strange behavior, has muscle rigidity, and a Glasgow Coma Scale score of 14. ABC is normal. Arterial blood gas shows P-Na = 118 mmol/L.		
**Correct fluid**		
3% NaCl, [Na^+^] = 513 mmol/L	59	(29.4)
**Inappropriate fluids**		
Isotonic saline, [Na^+^] = 154 mmol/L	76	(37.8)
0.9% NaCl with 5% glucose, [Na^+^] = 154 mmol/L	6	(3.0)
**Incorrect hypotonic fluids**		
Glucose 5% isotonic	3	(1.5)
Darrow-glucose, [Na^+^] = 31 mmol/L	0	(0)
Potassium-sodium-glucose, [Na^+^] = 40 mmol/L	12	(6.0)
0.45% sodium chloride with 2.5% glucose isotonic, [Na^+^] = 77 mmol/L	4	(2.0)
Ringer’s acetate, [Na^+^] = 130 mmol/L	14	(7.0)
**Do not know**	27	(13.4)

**Table 4 jcm-09-02790-t004:** Correct responses to knowledge questions *.

	*n*	(%)
Q10: Which of the following sentences are correct? (*n* = 198)	158	(79.8)
Most often acutely ill patients in need of IV fluids have increased renal water excretion **Most often acutely ill patients in need of IV fluids have decreased renal water excretion** Most often acutely ill patients in need of IV fluids have normal renal water excretion		
Q11a: How will Darrow-glucose ([Na^+^] = 31 mmol/L) affect the P-Na in a patient with decreased water excretion? (*n* = 184)	85	(46.2)
Large increase in P-Na with a risk of sodium overload Slight increase in P-Na Unchanged Slight decrease in P-Na **Large decrease in P-Na with a risk of hyponatriaemia**		
Q11b: How will Potassium-sodium-glucose ([Na^+^] = 40 mmol/L) affect the P-Na in a patient with decreased water excretion? (*n* = 179)	56	(31.3)
Large increase in P-Na with a risk of sodium overload Slight increase in P-Na Unchanged Slight decrease in P-Na **Large decrease in P-Na with a risk of hyponatriaemia**		
Q11c: How will Ringer’s lactate ([Na^+^] = 130 mmol/L) affect the P-Na in a patient with decreased water excretion? (*n* = 179)	43	(24.0)
Large increase in P-Na with a risk of sodium overload Slight increase in P-Na Unchanged **Slight decrease in P-Na** Large decrease in P-Na with a risk of hyponatriaemia		
Q11d: How will 0.9% NaCl with 5% glucose ([Na^+^] = 154 mmol/L) affect the P-Na in a patient with decreased water excretion? (*n* = 176)	55	(31.1)
Large increase in P-Na with a risk of sodium overload Slight increase in P-Na **Unchanged** Slight decrease in P-Na Large decrease in P-Na with a risk of hyponatriaemia		
Q12 (*n* = 173): In case of increased blood sugar (above 12 mmol/L), the measured plasma sodium (P-Na) must be corrected because the measured P-Na is:	52	(30.1)
**“Falsely low”** “Falsely high” There is no need for correction Do not know		
Q13: Which of the following diseases/symptoms may be indicative of potentially increased intracranial pressure (ICP)? (*n* = 164)	23	(14.0)
**Meningitis** Shortness of breath **Concussion** Chest pain **Seizure** **Acute liver failure** Acute abdomen Hip fracture		
Q14: Which of the following symptoms are indicative of severe symptoms of hyponatraemia and require acute treatment of hyponatraemia? (*n* = 163)	71	(43.6)
**Altered level of consciousness** **Seizure** Infection Chest pain **Muscle rigidity** Anaemia		
Q15a: What is the maximum correction of P-Na for a patient at high risk of osmotic demyelination? (*n* = 162)	54	(33.3)
**6 mmol/L** 8 mmol/L 16 mmol/L 20 mmol/L		
Q15b: What is the maximum correction of P-Na for a patient without high risk of osmotic demyelination? (*n* = 162)	48	(29.6)
6 mmol/L **8 mmol/L** 16 mmol/L 20 mmol/L		
Q16: How would you prevent plasma sodium from rising too rapidly and thus exceeding the recommended limits for P-Na correction? (*n* = 161)	62	(38.5)
**I record the administration of IV fluids** I recommend fluid restriction **I record the first low P-Na level** **I monitor P-Na regularly** I administer 5% glucose **I record the maximum recommended increase of P-Na** I encourage the patient to drink water		
Q17: Which of the following treatments would you initiate if a patient’s plasma sodium raises too rapidly? (*n* = 159)	89	(56.0)
**Water per os** Fluid restriction (fluid intake is limited to less than 1 L/day) Isotonic saline (1 L contains 9 g [154 mmol] sodium chloride) Ringer’s acetate (1 L contains 130 mmol sodium [4.1 g sodium acetate and 5.9 g sodium chloride], 0.295 g calcium chloride, 0.3 g potassium chloride, 0.2 g magnesium chloride) 3% NaCl (1 L contains 30 g [513 mmol] sodium chloride) **Glucose 5% isotonic (1 L contains 55 g [278 mmol] glucose)**		
Q18: What is the most common cause of over-correction? (*n* = 159)	14	(8.8)
**Large diuresis** Increased sodium secretion Increased water intake Inadequate water intake Increased sodium intake Reduced renal water excretion		

* Percentages of correct responses for knowledge questions are based on those who answered the specific question. The total number of respondents is indicated for each question in brackets. Correct responses are marked in bold. All questions included an option of ‘do not know’, which is not shown in the table.

**Table 5 jcm-09-02790-t005:** Critical prescribing practice, demographics, and characteristics of emergency departments (EDs) by simple logistic regression conducted separately for each individual predictor.

Variable	Prescribing Practice (Correct Responses)				Odd Ratio	95%CI	*p*-Value
	Critical (correct < 2) *n* (%)		Non-critical (correct ≥ 2) *n* (%)				
**Age**							
18–34 years	31	(38.8)	49	(61.3)	1.4	0.8–2.5	0.3
≥35 years	38	(31.4)	83	(68.6)	(ref *)		
**Number of weekly treated patients with intravenous fluids ****							
0–5 patients	36	(37.5)	60	(62.5)	1.4	0.8–2.4	0.3
>5 patients	32	(30.8)	72	(69.2)	(ref)		
**Years of practice ****							
≤5 years	29	(37.2)	49	(62.8)	1.2	0.6–2.2	0.6
>5 years	40	(33.1)	81	(66.9)	(ref)		
**Position *****							
Junior	39	(35.1)	72	(64.9)	1.1	0.6–1.9	0.8
Senior	30	(33.3)	60	(66.7)	(ref)		
**Size**							
Small/Medium	43	(35.2)	79	(64.8)	1.1	0.6–2.0	0.7
Large	26	(32.9)	53	(67.1)	(ref)		
**Complexity**							
Medium	52	(37.4)	87	(62.6)	1.6	0.8–3.0	0.2
High	17	(27.4)	45	(72.6)	(ref)		
**Type ******							
Pediatric ED	35	(34.7)	66	(65.3)	1.0	0.4–2.4	1.0
Adult ED	23	(33.3)	46	(66.7)	1.0	0.4–2.4	0.9
Combined general population ED *****	10	(34.5)	19	(65.5)	(ref)		

* Ref = reference. ** Missing answers were not included. *** Junior physicians include: foundation doctor year 1 and 2, specialty registrar and ‘other’ (medical students (7), unspecified junior doctor (4), pre-FY1 (2), and a PhD student). A senior doctor is a consultant. **** Trauma centers were excluded from this part of the analysis since only 2 respondents participated from this type. ***** Combined general population EDs provide care for all patients in one area, while separate general population EDs provide care to children and adults in separate locations within a facility.

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
