# Peer review of "Are Further Interventions Needed to Prevent and Manage Hospital-Acquired Hyponatraemia? A Nationwide Cross-Sectional Survey of IV Fluid Prescribing Practices"

_jcm, 2020, doi:10.3390/jcm9092790_

Round 1

Reviewer 1 Report

The authors surveyed ED physicians in Denmark to assess adequacy of prescribing practices for and fund of knowledge about hyponatremia. Use of a survey is a novel approach. 

The response rate was only 60% by ED and only 55% by ED physician, and the study was conducted only in Denmark. Together, the low response rate and restricted geographic distribution limit accuracy and generalizability of the study. The Discussion should much more strongly point out these shortcomings.

The questionnaire was not provided for review. I am unfamiliar with Ringer's acetate or Darrow glucose solution. Assuming Ringer's acetate has a similar sodium concentration to the Ringer's lactate available in the US, I doubt that use of Ringer's acetate as opposed to normal saline would have any deleterious effect on serum sodium response. Given the recent data favoring use of balanced, base-containing fluid for resuscitation rather than solutions containing chloride only, it may not be appropriate to count use of Ringer's acetate as an incorrect answer. Similarly, hypertonic saline would not be required in a patient with hyponatremia, moderate symptoms, and hypovolemia. In such patients, mere correction of hypovolemia would be accompanied by a water diuresis with spontaneous correction of hyponatremia. Over rapid correction often follows correction of volume depletion and would be even more likely with use of hypertonic saline. Clearly, use of hypotonic fluid is incorrect in this situation, but isotonic fluid may be appropriate, particularly since the patient was somnolent but not having a seizure. What exactly was described in this patient scenario and whether hypovolemia was contributory is unknown to me. The authors need to do a better job of convincing reviewers and readers that the authors have properly graded the questionnaires. As of now, it is not certain that the authors' grading scheme is correct. 

I doubt the location of EDs (Mid-Jutland vs Zealand) is of interest to an international readership.

In terms of distribution of cases (Table 3), the percentages should be calculated horizontally, not vertically. What is of interest is whether those in the 18-34 group were mostly good or poor prescribers not what percentage of poor prescribers were younger and what percentage were older. Do unsupervised 18 year-olds really have prescribing privileges in Danish EDs? Doesn't Junior vs Senior give the same information?

Did ED physicians whose practice is limited to adults do worse on the pediatric question?

Are there validated data showing what the most common cause of over-correction is? I doubt it. As such, the 8.8% correct answer rate on Q18 is not surprising.

The Introduction is overlong. Guidelines can be summarized rather than recounted one by one in detail. Similarly, the Discussion should focus on performance of ED physicians rather than on educating readers on the pathophysiology of hyponatremia. 

The authors may wish to cite Verbalis Am J Med 2013 126;10A;S1-S41 or Greenberg Kidney International 2015;88:167-177

Reviewer 2 Report

Re: Are further interventions needed to prevent hospital-acquired hyponatremia? A nationwide cross-sectional survey of IV fluid prescribing practices (Sindhal P t al.)

The report by Syndhal P et al. is potentially interesting (an the questionnaire appears well structured)

My comments (not presented in order of importance)

1) The section “Introduction” is really too long (this comment pertains, at least in part, the whole manuscript)

2) The main issue of the report is the prevention of hyponatremia. I wonder if the issue “management of symptomatic hyponatremia” should be included or not. Two options: a) exclude the latter issue or b) modify the “title” (e.g.: to prevent and manage hospital-acquired …)  and the philosophy of the communication.

3) The sodium level is often low both in children and in adults on admission

- In childhood, hyponatremia is common in cases with community-acquired infections such as bronchiolitis, acute diarrhea, and pyelonephritis (Mazzoni MB. PLoS One 2019;14:e0219299; Memoli E. Pediatr Nephrol 2020;35:713-714)

- In adulthood, hyponatremia is common in subjects managed with diuretics o blockers of the renin-angiotensin system (Arampatzis S. Am J Med 125:1125.e1-1125.e7)

4) Reference values from circulating sodium. The term normonatraemia has long been used to denote a sodium level ranging from 131 to 149 mmol/L. Most authorities (including laboratory reference values published in N Engl J Med and UpToDate) nowadays feel that sodium is normal from 135 to 145 mmol/L. Consequently, I was really surprised by the values suggested in this manuscript (135-142 mmol/L)

5) The authors should provide the time required (“estimated”) to answer the questionnaire (a possible explanation for comment number 5)

6) Participation in the study (= respondents) was rather low. This fact should be acknowledged (and discussed)

7) Who was invited to answer the questionnaire: general internists (N=?) and pediatricians (N=?)

8) Circulating sodium is nowadays measured either in plasma (or serum) by means of indirect potentiometry (= diluted sample) or in whole blood by means of direct potentiometry (= undiluted sample). The two techniques often present discordant results, as recently discussed among others in: Milani GP. Clin Chem Lab Med 2020;58:e117-e119). Regrettably, awareness of this discrepancy is lacking.

9) Line: 223 I do not understand the statement “Error! Reference source not found”

Reviewer 3 Report

Please refer to the attached file

Round 2

Reviewer 1 Report

Manuscript significantly improved. 

Selection of correct answers remains too rigid, particularly for scenario 2, where 41.9% were scored incorrect for omitting glucose, a solute that has nothing to do with hyponatremia, and scenario 4 where 37.8% were scored incorrect for administering isotonic fluid instead of hypertonic fluid in a situation at high risk for over rapid correction due to spontaneous water diuresis. Many authorities would argue that despite having modest symptoms of hyponatremia, the patient was at little risk of a seizure and that the safest course of action would be to administer isotonic fluid and observe the response of serum sodium. The latter course would make development of osmotic demyelination far less likely than if hypertonic saline were used. When only 1.5% of respondents get all the answers correct, one must criticize the test as much as the test takers. It would be interesting to see how the results changed if these two alternative answers were also scored as acceptable.

The manuscript would be strengthened outcomes were reported with more liberal grading in addition to the rigorous but debatable grading.

One typo: Europe, not Europa line 75.

Reviewer 2 Report

Excellent revision